# Molecular Structure of Salicylic Acid and Its Hydrates: A Rotational Spectroscopy Study

**DOI:** 10.3390/ijms25074074

**Published:** 2024-04-06

**Authors:** Alberto Macario, Juan Carlos López, Susana Blanco

**Affiliations:** 1Departamento de Química Física y Química Inorgánica, Facultad de Ciencias, IU CINQUIMA, Universidad de Valladolid, 47011 Valladolid, Spain; alberto.macario@uva.es (A.M.); juancarlos.lopeza@uva.es (J.C.L.); 2Département de Physique Moléculaire, IPR (Institut de Physique de Rennes), CNRS-UMP 6251, Université de Rennes, F-35000 Rennes, France

**Keywords:** molecular structure, water complexes, rotational spectroscopy, gas-phase, biological molecules and aggregates

## Abstract

We present a study of salicylic acid and its hydrates, with up to four water molecules, done by employing chirped-pulse Fourier transform microwave spectroscopy. We employed the spectral data set of the parent, ^13^C, and ^2^H isotopologues to determine the molecular structure and characterize the intra- and intermolecular interactions of salicylic acid and its monohydrate. Complementary theoretical calculations were done to support the analysis of the experimental results. For the monomer, we analyzed structural properties, such as the angular-group-induced bond alternation (AGIBA) effect. In the microsolvates, we analyzed their main structural features dominated by the interaction of water with the carboxylic acid group. This work contributes to seeding information on how water molecules accumulate around this group. Moreover, we discussed the role of cooperative effects further stabilizing the observed inter- and intramolecular hydrogen bond interactions.

## 1. Introduction

Salicylic acid (2-hydroxybenzoic acid, SA) is a widely used compound with applications in food science and pharmacology. It is a plant hormone that plays an important role in plant defense against stress through different mechanisms, having a significant impact on photosynthesis, transpiration, the uptake and transport of ions, or plant growth [1]. SA was first identified in willow (genus *Salix*) bark, which has been used since ancient times to alleviate pain and reduce fevers due to its analgesic, antipyretic, and anti-inflammatory properties [2]. Furthermore, SA is the main precursor of aspirin [3]. Nowadays, SA is widely employed in dermatology as a keratolytic and bacteriostatic agent [4] or for the treatment of acne, psoriasis, and other cutaneous diseases [5]. All these biological activities related to the physical and chemical properties of SA are defined by the molecular structure of this biomolecule.

The molecular structure of SA is governed by the *ortho* disposition of its two functional carboxyl and hydroxyl groups. The possible intramolecular interactions in *ortho* isomers confer them properties different from those corresponding to *meta* or *para* derivatives [6,7]. Moreover, in benzene derivatives, the presence of functional groups that are asymmetric, with respect to the substitution axis of the phenyl ring, is associated with an alteration of the ring structure and the electronic aromatic behavior. This is due to the stabilization of one of the canonical forms of the aromatic ring over the other, a phenomenon known as the angular-group-induced bond alternation (AGIBA) effect [8,9]. The knowledge of the molecular structure of SA and of the interaction between its two groups in *ortho* disposition have been the subject of different investigations [10,11,12,13,14,15]. Two conformers of SA, I and II (see Figure 1), were reported to exist from a study of the jet-cooled IR-UV double-resonance spectrum [10]. By contrast, only the global minimum I form was identified from the free jet millimeter-wave absorption spectrum [11]. Both I and II conformers present planar structures with an intramolecular hydrogen bond (HB) O-H⋯O from the hydroxyl group to the carbonyl (I) or the hydroxyl (II) moieties of the carboxylic acid group. The molecular structure of SA-I has been determined by electron diffraction [12], suggesting that the intramolecular HB interaction is further stabilized by resonance-assisted hydrogen bonding (RAHB) [16]. In other studies, SA has also been employed as a model of the investigation of keto-enol tautomerism observed by excited-state intramolecular proton transfer (ESIPT) [13,14,15].

Investigation of microsolvated molecular systems is a relevant subject in chemistry and biochemistry [17,18,19]. Microwave spectroscopy techniques are exceptional tools to determine their structures, adequate for this purpose due to their inherent high resolution [20]. There are a high number of microwave studies of microsolvated organic molecules [19,21,22,23,24,25], providing models for a better understanding of the water interactions with biomolecules [26], of hydrogen bond (HB) cooperativity [27], and of the way in which solvation induces structural changes in the solute molecule [22,23,27,28]. These studies have also led to an understanding of the role of association processes and the interplay between the self-association of water and solvation. With few exceptions [22], in complexes with several H_2_O molecules, water prefers to link other water molecules, forming chains or cycles. When the solute has only one HB acceptor site, the structures reflect a balance between maximizing the number of water-solute interactions and the minimum-energy structures of the pure water clusters [21]. In solutes with double HB donor/acceptor character, water molecules close sequential cycles, as observed in amides [28], acids [29,30,31,32], or esters [33]. In the last years, studies of molecules forming clusters with a high number of water molecules has increased. The microsolvation of benzene derivatives is also of interest [34], since they are archetypal molecules in chemistry, and their water complexes serve as model chromophores to investigate solute-solvent interactions. Understanding these interactions in aromatic compounds is fundamentally important for studying more complex systems encountered in many natural chemical reactions [34]. In the specific case of SA, the study of its microsolvates is particularly relevant and of interest in atmospheric chemistry [35].

In a recent study [36], we observed the presence of SA as a thermal recombination product of *o*-anisic acid, together with methyl salicylate [37], methyl 2-methoxybenzoate [38], and their complexes with water [38,39]. Water could interact with SA in multiple ways due to the presence of several donor or acceptor groups in this molecule. As inferred from gas-phase microsolvation studies of related acids [29,31,40,41], the most favorable interaction sites correspond to the carboxyl group, with which water molecules may easily close cycles. Another aspect to consider is how microsolvation affects intramolecular interactions. A special situation is observed in the monohydrate of *o*-anisic acid [36], where each of the two observed conformers of the monomer forms its corresponding water cluster. One complex maintains an intramolecular interaction with a *trans*-COOH arrangement, and the other presents a *cis* carboxylic acid disposition that establishes the typical sequential ring of the carboxyl group with water.

In this work, we have taken advantage of the potential of the chirped-pulse Fourier transform microwave spectroscopy (CP-FTMW) technique, aided by supersonic jets, to study SA microsolvates with multiple water molecules. We present the gas-phase molecular structure of SA, determined through extensive isotopic species measurements. We have experimentally analyzed the structural properties of SA, such as the AGIBA effect [8,9] and the intramolecular HB interaction. We have observed and analyzed the spectra of several SA hydrates with up to four water molecules to gain information on how water molecules accumulate around an acid group, as well as information on the structures adopted by these clusters. Another aspect on which we have focused our work is the identification of pieces of evidence of cooperative effects that further stabilize HB intramolecular interactions in the complexes [42] as the number of associated interacting molecules increases.

## 2. Results and Discussion

### 2.1. Rotational Spectrum

To address the rotational spectrum of SA (see Figure 2 and Appendix A), we used the rotational parameters determined in the millimeter-wave study [11] to predict its rotational spectrum in the 2–8 GHz microwave region. We measured a new set of 64 *µ*_b_-type R- and Q–branch transitions, with J ranging from 1 to 10. The analysis of the spectrum was conducted using Watson’s semirigid rotor Hamiltonian in the A reduction and the I^r^ representation [43], including the transitions previously observed [11]. The resulting rotational parameters are compared with those previously obtained from millimeter-wave data [11] and with the theoretical predictions in Table 1. The high S/N ratio allowed us to identify the spectra of all the ^13^C isotopologues in their natural abundances, as shown in Appendix A. Moreover, when we included deuterium oxide (D_2_O) into the reservoir adapted to the carrier gas line, the deuterium atoms substituted the hydroxyl and carboxyl hydrogen atoms. As a result, we observed the spectra of the mono- and disubstituted deuterated species. The analyses of the spectra of the isotopologues were conducted using the same Hamiltonian used for the parent, with the quartic centrifugal distortion constants fixed to the values determined for the parent species. All the rotational parameters determined are listed in Appendix A. No other SA conformations were identified in the rotational spectrum in agreement with the calculated relative energies and the predicted values of the dipole moments (Figure 1 and Appendix A).

Once the transitions observed for the monomer and its isotopologues were removed, we were able to identify the spectrum of a new species with a set of over 60 *µ*_a_-type R-branch and *µ*_b_-type R- and Q–branch transitions. No *µ*_c_-type lines were observed. These lines were attributed to a monohydrated SA-w species (Figure 2). The analysis of this spectrum gave the set of rotational parameters listed in Table 2. A conformational search of the monohydrated clusters for the two most stable conformers of SA was performed theoretically by exploring the possible interactions of water and SA, both with a dual proton donor/proton acceptor character (Appendix A). The values of the experimental rotational constants are close to those predicted for the two lower energy conformers of the monohydrated cluster, I-w-a and II-w-a (see Figure 1 and Table 2). There are two reasons to consider conformer I-w-a as the observed species. Firstly, species I is the only one observed in the spectra for the SA monomer, so it will be the only species present in the molecular jet to form any cluster [36]. Secondly, the monohydrated cluster I-w-a is predicted to be the global minimum, being 900 cm^−1^ more stable than conformer II-w-a (Figure 1). This assignment was unambiguously corroborated by the measurements of monodeuterated isotopologues and the subsequent study of the monohydrate molecular structure (see below). No other monohydrated forms of SA were identified.

To explore the complexes of SA with a higher number of water molecules, conformational searches were conducted for SA-w_2_, SA-w_3_, and SA-w_4_ clusters using the CREST (Conformer-Rotamer Ensemble Sampling Tool) application [44] followed by B3LYP-D3/6-311++G(d,p) [45,46,47,48] structure optimizations. Some of the resulting structures are given in Appendix A, and the spectroscopic parameters are given in Appendix A. From the predicted energies, it can be deduced that the predicted preferred interaction sites are related to the COOH group. As examples, we have included some of the possible complexes with water molecules closing rings with the phenol group that have, in all cases, rather high energies relative to the global minima.

In the next step, we identified a weaker spectrum consisting of over 125 *µ*_a_-type R-branch and *µ*_b_-type R- and Q–branch transitions. These transitions were analyzed using the same Hamiltonian as before, and a set of rotational constants related to a dihydrated cluster SA-w_2_ was determined (Table 2). Due to its weak intensity, the deuterated species could not be identified in the spectra enriched with deuterium oxide. As with the monohydrated species, the set of rotational constants is comparable to those predicted for the I-w_2_-a and II-w_2_-a species (see Table 2). Following the same arguments considered for the monohydrated complex, the dihydrated species was identified as the predicted global minimum I-w_2_-a conformer.

After removing all the measured lines of the SA-w and SA-w_2_ complexes, there was still a dense spectrum of weaker lines. Searches for the spectra of the SA-w_3_ cluster led to the identification of a series of doublets attributable to one of such complexes. Most of the *µ*_a_-type R-branch and *µ*_b_-type R- and Q-branch transitions were split by 50–200 kHz with doublets of the same intensity (see Figure 2). This feature has been observed in similar trihydrates such as formamide-w_3_ [28], ethyl carbamate-w_3_ [49], or methyl carbamate-w_3_ [50], where the chain of three water molecules closes a cycle with the amide group. Considering the four most stable conformers that present this water trimer disposition with the acid group of SA, only two have rotational constants close to the experimental ones (Table 3 and Appendix A). Nevertheless, the observed conformer was assigned to the global minimum I-w_3_-a, since the observed transitions are only in agreement with the selection rules derived from the values of the predicted dipole moment components for this conformer. A two-states Hamiltonian, in this case using Watson’s semirigid S reduction [43], including Coriolis coupling terms [51], was used to analyze the spectrum. The centrifugal distortion constants were kept equal for both states. It was possible to determine the Coriolis coupling term and the energy difference between both vibrational states. The results are summarized in Table 3.

Finally, with the remaining lines, we were able to assign another spectrum attributable to an SA-w_4_ cluster. The spectrum is composed of over 170 *µ*_a_- and *µ*_b_-type transitions. By comparing the values of the rotational constants and the observed selection rules with the theoretical values and the predicted dipole moments (Appendix A), the observed species can be identified with one of the two most stable conformers. We tentatively assigned the observed species to the global minimum I-w_4_-a (Appendix A). A summary of the results is shown in Table 4.

All the measured transitions (Appendix A) and rotational determined parameters (Appendix A) for SA and its hydrated clusters are given in the Appendix A.

### 2.2. Molecular Structure

The degree of planarity of a given species is typically discussed in terms of the value of the planar moment of inertia *P*_cc_, which represents the mass extension out of the *ab* inertial plane. As pointed out in the previous study [11], for an SA monomer, this value is close to zero for the parent species and remains at practically the same values for all observed isotopologues, indicating that all the substituted C atoms, as well as the hydroxyl and carboxyl hydrogen atoms, lie in the *ab* inertial plane. The small value of *P_cc_* (0.12962(19) uÅ^2^) could be attributed to vibrational contributions [52]. Moreover, this value is smaller than that obtained for benzoic acid (0.183 uÅ^2^) [53] and larger than for phenol (0.015 uÅ^2^) [54]. These differences can be explained in terms of the intramolecular HB interaction in SA, which limits the out-of-plane vibrations of the carboxyl group in SA compared to benzoic acid, demonstrating that the presence of an intramolecular interaction somewhat decreases the flexibility of the molecule. The SA-w complex presents a slightly higher *P*_cc_ value (0.51743(24) uÅ^2^), attributable to the preferred orientation of water interacting with the carboxyl group that leaves the non-interacting H_19_ atom out of plane, and to possible intermolecular vibrational contributions. Moreover, this value is coherent with a planar skeleton structure. This structure is consistent with the *P*_cc_ values of the ^2^H isotopologues, except for that of the H_19_ atom, corroborating the conclusion of a planar skeleton structure of an SA-I-w-a complex. The dihydrated SA-I-w_2_-a conformer presents a value of *P*_cc_ (1.38688(57) uÅ^2^), consistent with two non-interacting water hydrogen atoms out of plane. The planar moment is then compatible with a planar skeleton structure, including both the water oxygen and the interacting hydrogen atoms lying in the *ab* inertial plane. The value of *P*_cc_ for the trihydrated complex (9.9634(11) uÅ^2^) reflects the contribution of water molecules lying slightly out of the *ab* plane, as occurs in other related complexes such as formamide–w_3_ [28], ethyl cabamate-w_3_ [49], or methyl carbamate–w_3_ [50]. The same behavior is observed for the SA-w_4_ complex with a value of *P*_cc_ (113.4455(21) uÅ^2^), much higher than the other values observed in this work, indicating that the configuration of this complex is nonplanar.

The complete sets of experimentally determined rotational constants for the parent species of SA and SA-w and their isotopologues were employed to determine the substitution *r*_s_ [55,56] and effective *r*_0_ [57] structures. The substitution structure employs the Kraitchman method [55] to derive the absolute values of the coordinates of every isotopically substituted atom in the principal inertial axis system of the parent species. The method relies on the variation of the experimental moments of inertia upon substitution. The Costain formula [56] is usually used to estimate coordinate uncertainties. The sign of the obtained coordinates had to be resolved from a reasonable structure. Nevertheless, this method may have problems for atoms lying close to the inertial axes due to zero-point vibrational effects. On the other hand, the effective structure is determined by a least squares fit of all the experimentally determined rotational constants [57]. However, for the SA-I conformer, the effective *r*_0_ structure could not be well determined. That was also the case for planar species such as *o*-anisic acid [36] and anisole [58], due to the effects of vibrational contributions. Following previous works [36], the mass-dependent *r*_m_ structure [59] was complementarily employed in this case. This structure introduces into the least squares fit a limited set of additional adjustable parameters that consider the mass-dependence of the vibrational contribution to determine equilibrium quality parameters [59,60]. For the SA-w monohydrated species, the *r*_m_ parameters obtained for the monomer were kept fixed to determine its effective structure. Finally, for the remaining hydrated species (SA-w_2_, SA-w_3_, and SA-w_4_), as no isotopologues could be identified in the spectra, the equilibrium *r*_e_ structures were considered as a good estimation of the molecular geometries, owing to the good agreement between the predicted and the experimental rotational constants.

#### 2.2.1. Monomer (SA-I)

The experimental *r*_s_ and *r*_m_ structures, the gas-phase electron diffraction *r*_a_ structure [12], and the equilibrium *r*_e_ structure are compared in Appendix A. Figure 3a compares the *r*_s_ and *r*_e_ structures. Overall, there is a good agreement between the different methods for all geometrical parameters, except for the C_1_ atom, for which the *a r*_s_ coordinate results in an imaginary value, due to its proximity to the center of mass of the molecule.

It is worth analyzing the structural features associated to the O⋯H-O HB interaction between the hydroxyl and carboxyl groups, which closes a six-membered planar ring, comparing SA with phenol [54,61] and benzoic acid [12]. The *r*_m_ (O_12_-H_16_) distance (0.990(14) Å) increases when compared to phenol (*r*_s_ = 0.957(6) Å [54], *r*_a_ = 0.958(3) Å [61]) (see Figure 3 for atom labelling). On the other hand, the *r*_m_ (C_2_-O_12_) length (1.342(13) Å) decreases (*r*_s_ = 1.375(5) Å [54]; *r*_a_ = 1.379(2) Å [61]). Comparing the structure of the carboxyl group with that of benzoic acid [12], the following changes are observed: the C_11_-O_13_ length decreases (1.3496(42) Å vs. 1.367(8) Å), the C_11_=O_14_ distance remains unchanged (1.2258(42) Å vs. 1.225(6) Å), and the C_1_-C_11_ distance decreases (1.4682(83) Å vs. 1.484(6) Å). The distance O_14_⋯H_16_ = 1.7569(40) Å is typical of a moderate HB, and thus the enlargement of the O_12_-H_16_ distance is a natural consequence of this interaction. The alteration of the rest of the parameters relative to benzoic acid and phenol could be attributed to the effects of a resonance-assisted hydrogen bond (RAHB) [16]. From the determined bond lengths and the predicted bond orders (see Figure 3b,c), C_3_-C_4_ and C_5_-C_6_ bonds exhibit a higher double-bond character compared to their C_2_-C_3_, C_4_-C_5_, and C_1_-C_6_ neighbor bonds. Thus, the structure of the phenyl ring suggests the prevalence of one of the aromatic canonical benzene forms relative to the other, as it is schematically shown in Figure 3b,c. This prevalence is consistent with the presence of an AGIBA effect governed by the hydroxyl group [8,9]. The fact that the C_1_-C_2_ bond may not be consistent with this effect is probably due to the intramolecular hydrogen bond RAHB effects. This competition between the AGIBA and the RAHB effects, which was also observed in *o*-anisic acid [36], influences the structure of the phenyl ring and contributes to understanding how these effects coexist. NBO calculations support these conclusions (see Appendix A).

#### 2.2.2. Monohydrated Complex (SA-w)

The *r*_s_, *r*_0,_ and *r*_e_ geometric parameters of the monohydrated complex of SA are compared in Appendix A and in Figure 4, where it is possible to observe a good agreement between them. The effective structure was calculated by keeping the phenyl ring *r*_m_ structure of the monomer fixed in the fit to determine the intra- and intermolecular interactions. The comparison between the monomer and heterodimer *r*_e_ structures, together with the predicted NBO conjugative stabilization energies (Appendix A), indicates that the ring structure is not substantially altered by the presence of water. In this complex water closes a six-membered sequential cycle with the carboxyl group through two O-H⋯O hydrogen bonds further stabilized by RAHB, as reflected by the structural parameters and calculated bond orders (Appendix A, respectively). This sequential cycle is also observed in the benzoic acid-water cluster [40], and in the monohydrated complexes of other benzene derivatives, such as the *o*-anisic acid [36] or the *p*-toluic acid [62], or other structures presenting a carboxylic acid such as formic acid [41], acetic acid [31], propanoic acid [29], or the di- and trifluoroacetic acids [30,32].

The dominant intermolecular interaction is the OH⋯O_w_ HB, reflecting the enhanced acidity of the carboxyl hydrogen atom (see Figure 4). The corresponding distance of 1.7531(24) Å (*r*_e_ = 1.764 Å, Appendix A) is shorter than that estimated for the benzoic acid-water complex (1.80 Å) [40]. The C=O⋯H_w_ HB distance is 1.9819(25) Å (*r*_e_ = 2.031 Å), larger than that reported for the benzoic acid-water heterodimer (1.95 Å) [40]. This could be explained by considering that the carbonyl moiety also supports the intramolecular HB with the *ortho* hydroxy group, sharing its electron density between both HBs. This, in turn, further stabilizes the dominant intermolecular OH⋯O_w_ HB interaction.

The formation of the OH⋯O_w_ HB naturally enlarges the carboxyl O-H distance (*r*_0_ = 1.0065(13) Å) relative to that of the SA monomer (*r*_m_ = 0.969(2) Å), as happens for the benzoic acid-water complex [40]. Even though the distances between C_11_ and both oxygen atoms could not be experimentally determined, the calculated equilibrium geometries show a shortening in the C-OH bond length and an enlargement of the C=O distance upon formation of the complex. These effects, due to RAHB, are evidenced by the changes in the calculated NBO bond orders and the delocalization energies when comparing the monomer with the monohydrated form (see Appendix A). The formation of the hydrated complex also seems to affect the geometry of the *ortho* alcohol group. The intramolecular HB is shorter than that observed for the monomer form, while there is an increase in the O-H bond length (see Appendix A). In other words, the interaction of water with the acid group appears to increase the strength of the intramolecular HB.

On the other hand, since the monohydrated I-w-a form presents a planar skeleton, we employed the method given by Ouyang and Howard [31] to determine the angle *θ* between the *a* principal axis of the monomer form and the intermolecular axis of the monohydrated cluster (see Appendix A). As a result, the calculated *θ* angle for I-w-a species is 7.7°, close to the observed for the water complex of benzoic acid (5°) [40]. Therefore, the intermolecular bonding axis can be considered to be nearly parallel to the *a* principal inertial axis of the observed monomer species.

#### 2.2.3. Dihydrated Complex (SA-w_2_)

For the dihydrated SA-w_2_ species, the non-observation of the spectra of any monosubstituted isotopologue precludes the determination of the *r*_0_ and *r*_s_ structures. However, the good agreement between the experimental and predicted rotational parameters, which is less than 1%, allows us to consider the *r*_e_ structure predicted for conformer I-w_2_-a as a reliable estimate of the molecular geometry (see Figure 4 and Appendix A). In this complex, a dimer of water closes a sequential eight-membered cycle with the carboxylic acid group through a chain of three O-H⋯O HBs. The intramolecular O-H⋯O interaction between the hydroxyl and carbonyl groups is also maintained. This arrangement is similar to other cases of acid dihydrates such as formic acid [41], acetic acid [31], propanoic acid [29], or di- and trifluoroacetic acids [30,32]. In all these cases where the water dimer is closing a cycle with a molecule bearing two closely positioned proton donor and proton acceptor groups, the water molecules lie in the molecular plane or in the plane of the carboxyl group, with the non-interacting hydrogen atoms out of the plane, pointing each one in opposite directions.

In this structure, the water dimer has a distance between the interacting hydrogen and oxygen atoms closer to the equivalent length of the carboxyl group, so the interaction between them presents a natural arrangement less forced than the monohydrated conformer. The hydrogen bond angles are close to the canonical value of 180° (see Appendix A). Cooperative forces are favored in homodromic cycles, reinforcing the HB interactions, leading to a shortening of the hydrogen bonds when comparing the HB distances with those of the monohydrate, as shown in Figure 4. As pointed out previously [28], the O_w1_⋯O_w2_ distance (2.682 Å) is shorter than that reported for water dimer (2.976 Å) [64], and even shorter than the O_w_⋯O_w_ distance (2.78 Å) of the water tetramer [65]. All these O⋯O distances related to hydrogen bonds are listed in Table 5.

#### 2.2.4. Trihydrated Complex (SA-w_3_)

The complex assigned to the I-w_3_-a conformer (Figure 1) shows a chain of three water molecules closing a 10-membered ring with the carboxyl group. Compared to the dihydrated complex, the presence of the third water molecule alters the planarity and the dynamics of the molecule. In terms of planarity, the oxygen atom (O_w1_), which interacts with the OH-carboxyl moiety (see Figure 4), is close to the SA plane, slightly inclined backward (−4.3°, with respect to the SA *ab* plane), while the other water molecules lie out of plane. Thus, O_w2_ is tilted over 25° and O_w3_ around 10° ahead of the *ab* plane of the monomer (see Appendix A). This is consistent with the experimentally observed *P*_cc_ value. This deviation from planarity contrasts with other acid trihydrates, such as di- and trifluoroacetic acids [30,32], which are closer to planarity. Similarly, the planar molecule of formamide forms a complex with three water molecules, presenting a *P*_cc_ value of 7.03654(6) uÅ^2^ [28], close to the value observed for SA-w_3_ (9.963(1) uÅ^2^, Table 3). Moreover, the most stable conformations for SA-w_3_ (Appendix A) closely resemble those found for the formamide-w_3_ complex. Regarding the dynamics of the molecule, the observation of tunneling doublets suggests an inversion of the configuration, similar to what occurs in formamide-w_3_ [28]. In this study, a path for the inversion of the configuration alternative to that passing by a high-energy planar molecule transition configuration was identified through successive flipping motions of the water molecules. As shown in Appendix A, a similar path is calculated for SA-w_3_. This periodic pseudorotation function with reasonably low barriers could explain the observed doublets associated with the dynamics created by the presence of the third water molecule.

Based on the predicted equilibrium structure (Appendix A) and the NBO calculations (Appendix A), it appears that the structure of SA, particularly the CO-H and C=O acid structure and the intramolecular HB, is not significantly affected by the presence of the third water molecule. On the other hand, the cooperative effects result in a shortening of all intermolecular HB distances between water molecules and SA compared to SA-w_2_ and SA-w complexes, as depicted in Figure 4. Examining the distances between the oxygen atoms involved in the HBs (Table 5) reveals a shortening in the distances associated with the HBs between SA and the water molecules (O_13_⋯O_w1_ and O_14_⋯O_w2_), which is pronounced in the OH⋯O_w1_ HB.

#### 2.2.5. Tetrahydrated Complex (SA-w_4_)

In the SA-w_4_ complex assigned to I-w_4_-a (Figure 1 and Figure 4), the trend observed for the SA-w_1_, SA-w_2_, and SA-w_3_, where a chain of water molecules closes a cycle with the carboxyl group, breaks down. In this case, it appears as if the SA-w_2_ complex has captured a second water dimer. The two water molecules interacting with SA in SA-w_4_ have a similar arrangement as that for SA-w_2_. This is further corroborated after removing the contribution of the third and fourth water molecules, obtaining the same structure and planar moments as for the SA-w_2_. The equilibrium structure (see Appendix A) also confirms that the oxygen atoms of these two water molecules interacting with the carboxyl group are in the SA plane. As mentioned, the SA-w_2_ has a favorable geometry for interaction with the carboxyl group. The second water dimer closes a water tetramer cycle in the most favorable arrangement. A similar disposition has been observed for the formamide-w_4_ complex [67]. In this case, the plane of the water tetramer is bent over 110° (see Appendix A), with respect to the plane of the SA-w_2_ moiety.

In this cluster, as in the previous cases, cooperative effects shorten the HB distances between water and SA. In Figure 4, it can be observed that these distances progressively decrease upon the addition of new water molecules. However, in SA-w_4_, the water molecules do not form a sequential cycle with the carboxylic acid group. Consequentially, the HB distances between water molecules notably increase, diverging from the trend observed by the SA-w_2_ and SA-w_3_ clusters. However, the intramolecular HB is also shortened compared to the other species, indicating further stabilization of cooperativity between the four water molecules and SA. All these pieces of evidence were also corroborated by the NBO analysis (see Appendix A).

### 2.3. Cooperative Effects and Dissociation Energies

When discussing HB cooperativity, two aspects are usually considered. One of them involves HBs between molecules with multiple conjugated π bonds (known as RAHB or π-cooperativity) [16,42,68]. An example is the reinforcement of the O-H⋯O intramolecular HB in SA. The second aspect is related to the formation of linear or cyclic chains involving multiple molecules with double roles as HB donors and acceptors. We can consider the hydrates of SA observed in this work as good examples of σ cooperativity. A first consequence of cooperativity is the shortening of the hydrogen bonds, as described in the previous section (Figure 4). In the same way, an increase in the HB energy per bond should be analyzed.

A preliminary approach to analyzing the energies per bond involves calculating the dissociation energies. These have been obtained at MP2/6-311++G(2df,2dp) level using the counterpoise method to correct the energies for the basis set superposition error (BSSE) [69]. We selected this computational level to facilitate comparison with those obtained previously for the benzoic acid—water complex [40]. The positive value of the interaction energy obtained after BSSE correction represents the equilibrium dissociation energy *D*_e_. The dissociation energies (*D*_e_) are given in Table 6, together with the average energy per bond. The predicted *D*_e_ value in SA-w is 41.3 kJ/mol, which is close to the 41.9 kJ/mol obtained for the monohydrated cluster of benzoic acid. Hence, the presence of an *ortho* intramolecular HB does not seem to affect the dissociation energy. It is possible to observe that the average energy per bond increases as the number of water molecules increases along the series SA-w, SA-w_2_, SA-w_3_. However, it decreases for the SA-w_4_ complex consistently with the fact that this cluster is no longer formed through a single cycle as the smaller size clusters. In any case, the energy per bond in the SA-w_4_ cluster is higher than that calculated for the SA-w heterodimer. The reinforcement of the dissociation energy can be also interpreted as a signature of the σ cooperative effects when the number of water molecules increases in the clusters, and it is especially pronounced when the cluster consists of a cycle, as occurs for SA-w, SA-w_2_, and SA-w_3_ complexes.

To individually analyze the strength of each HB, we have performed QTAIM (Quantum Theory of Atoms in Molecules) [70,71] and NCI (Non-Covalent Interaction) [72] calculations. QTAIM analyzes the topology of the electron density to determine bond paths (BP) and bond critical points (BCP), while NCI analysis allows the visualization of the electron density regions associated with non-covalent interactions. Appendix A show the combined results of these analyses for all the identified SA species. Moreover, using the electronic density provided by the QTAIM analysis at the BCPs, the interaction energies of the HB can be estimated following the empirical equation proposed by Emamian et al. [63]. The interaction energy estimated for the intramolecular HB of SA slightly increases stepwise along the series from the monomer (34.24 kJ/mol) to the tetrahydrated complex (37.38 kJ/mol) (see Figure 4). The energy of the interaction between the carboxylic OH and the interacting water molecule (O_w1_) increases its value by approximately 12 kJ/mol from the mono- to the dihydrated cluster, maintains almost the same value in the trihydrate and experiences an increment close to 12 kJ/mol when passing to the tetrahydrated complex. The high value of the energy of this bond and its increment in energy are probably due to the strong acid character of the OH carboxyl proton. The same behavior is reflected in the HB interaction between the C=O of the carboxyl moiety and its water partner (O_w2_). It is worth mentioning that the intramolecular HB is dominant in SA and SA-w, while in the other clusters, the carboxyl OH–O_w1_ HB is dominant.

## 3. Materials and Methods

### 3.1. Experimental Details

The rotational spectra of SA and its SA hydrates were investigated using a broadband chirped-pulse Fourier transform microwave spectrometer (CP-FTMW), as described elsewhere [73]. A commercial sample of SA (m.p. ~158 °C, b.p. ~211 °C), used without further purification, was placed in a reservoir at the heatable nozzle, where it was kept at 150 °C. A water reservoir inserted in the carrier gas line was used to increase the amount of water vapor in the expansion. Ne was used as the carrier gas, with a stagnation pressure of 2 bar and pulses of 700 µs. The gas mixture expanded supersonically into the vacuum chamber through a 0.8 mm nozzle. After a small delay, an arbitrary waveform generator created a 2–8 GHz chirped-pulse of 4 µs duration, which was then amplified to 200 W and broadcasted inside the chamber through a horn antenna, arranged perpendicular to the molecular expansion. The molecular transient emission signal was detected through a second horn antenna, preamplified, recorded with a digital oscilloscope (40 µs gate length), and Fourier-transformed into the frequency domain. For each molecular pulse, the polarization/detection steps were repeated 8 times. This operation sequence was repeated as soon as the optimum vacuum conditions were restored in the chamber, typically operating at a 5 Hz repetition rate. The accuracy of the frequency measurements is estimated to be better than 15 kHz. All the single substituted ^13^C isotopomers were measured in their natural abundance. To record the spectra of the deuterated isotopologues, a 1:1 mixture of water and deuterium oxide was placed in the reservoir. To measure and analyze the spectra, several available programs were used [74,75,76,77].

### 3.2. Theoretical Methodology

To explore the conformational landscapes of SA with two, three, and four water molecules, we employed the Conformer-Rotamer Ensemble Sampling (CREST) tool based on the xtb semiempirical extended tight-binding program package [44]. The resulting conformers were further optimized using the B3LYP hybrid density functional [45,46,47,48] with the D3 Grime’s dispersion correction term [78] and the Pople’s 6-311++G(d,p) basis set [79]. In addition to the three conformers predicted in previous work [11], for the monomer forms, we predicted two additional new stable forms, IV and V (see Figure 1 and Appendix A). The results for the hydrated complexes are compiled in Appendix A and Appendix A. For all the predicted species, harmonic frequency calculations were conducted to ensure that all the calculated conformers are true minima. Natural Bond Orbital (NBO) [80], non-covalent interactions (NCI) [72], and Quantum Theory of Atoms in Molecules (QTAIM) [70,71] analyses were performed for all the experimentally observed species at the same DFT B3LYP-D3/6-311++G(d,p) level. Complementary MP2/6-311++G(2df,2pd) [81] calculations were used to estimate the dissociation energies, employing the counterpoise method to correct the basis set superposition error (BSSE) [69]. All the DFT and MP2 calculations were done using the Gaussian 16 program package [82].

## 4. Conclusions

In this work, we have recorded the rotational spectra of SA, its monosubstituted ^13^C isotopologues, several ^2^H species, and the SA-w_n_ (n = 1–4) hydrates using CP-FTMW spectroscopy. The analysis of the experimental data has been complemented with computational chemistry calculations, including NBO, NCI, and QTAIM analyses, to gain a better understanding of the structural behavior of SA and its hydrates and to characterize the different inter- and intramolecular HB interactions established in these species.

The experimentally determined SA structure has allowed us to characterize the O-H⋯O intramolecular HB between the carboxylic acid and the hydroxyl functional group in *ortho* position. This HB forms a sequential six-membered cycle further stabilized by π cooperativity (RAHB), which reinforces the planarity and rigidity of the molecule. The AGIBA effect governed by the alcohol group is reflected in the benzene ring bond lengths. However, in proximity to the carboxylic acid or hydroxyl groups, the structure of the ring appears to be dominated by the RAHB effects.

The analysis of the spectra of the hydrated complexes revealed interesting insights into how water aggregates around an acid group. As we have already mentioned, the possible aggregates around the phenolic OH group all have high energies (see Appendix A). In the complex phenol-w, the O-H group acts preferably as a proton donor. The corresponding structure is predicted to be much more stable than the form in which water behaves as the proton donor [83]. However, in the most stable forms of SA, the proton donor capacity of the phenolic OH group is employed in the intramolecular interaction, so the possible SA-water complexes with water interacting with the phenol O-H group are expected to have higher energies, as it is confirmed by the calculations.

In the mono-, di-, and trihydrated clusters, water, or its dimer or trimer, forms a chain, closing a sequential cycle with the two HB donor OH and acceptor C=O ends of the carboxyl group. In the tetrahydrated species, water molecules form a tetramer cycle, bonding to the SA carboxyl group with contacts similar to those established in the SA-w_2_ cluster. While the mono- and dihydrated clusters maintain the planarity of the heavy atom skeleton of the complex, the spectra of SA-w_3_ and SA-w_4_ evidence of their non-planarity, as shown by the planar moment values *P*_cc_, further corroborated by the theoretical computations. The trihydrated cluster has two of the water molecules slightly out of the plane. In the SA-w_4_ complex, two of the water molecules are in the plane of SA bonded to the carboxylic acid group, keeping the form of SA-w_2_ species, while the other two lie above the plane of SA in an arrangement where the plane of the water molecules presents an angle of about 110° with respect to the SA-w_2_ plane. The cyclic structures of the hydrates of SA are comparable with those reported by Howard and coworkers on organic acid mono-, di-, and trihydrates [29,30,31,32]. However, in contrast with the non-planar heavy atom skeleton observed here for the trihydrate of SA, the trihydrates of di- and trifluoroacetic acids seems to have nearly planar heavy atom skeletons.

In the work on formamide-(H_2-_O)_3_ complexes [28], the structural relationship between the formamide-(H_2_O)_n_ clusters and the pure water clusters (H_2_O)_n+2_ was pointed out. The structures determined in this work for the SA-w_n_ complexes could be related in the same way to the structures determined or predicted for the (H_2_O)_n+2_ clusters [20,64,65,66,84]. The pieces of evidence found of the enhanced hydrogen bonding (HB), due to the cooperativity associated with the increased number of water molecules in the clusters, are remarkable. The evolution of the HB features reflected in the changes of the O··H and O⋯O distances, the BSSE corrected dissociation energies per HB, the HB strengths estimated from the QTAIM analysis, and the stabilizing delocalization energies predicted by the NBO calculations evidence the existence of σ cooperativity. The differences between the intermolecular HBs of I-w-a species and those of the monohydrated species of benzoic acid [40] indicate the strong influence of the intramolecular interaction altering the proton acceptor properties of the C=O functional group. In conclusion, the interaction with the water molecules in the clusters increases the strength of the intramolecular HB, which is the dominant interaction in complex

## Figures and Tables

**Figure 1 ijms-25-04074-f001:**
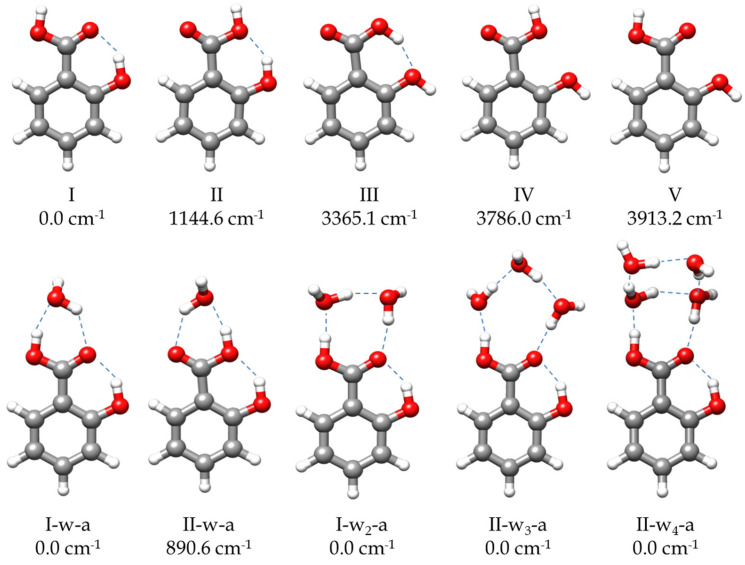
The most stable predicted forms of SA monomer and its monohydrated, dihydrated, trihydrated, and tetrahydrated complexes. The relative energies are calculated at B3LYP-D3/6-311++G(d,p) level of theory.

**Figure 2 ijms-25-04074-f002:**
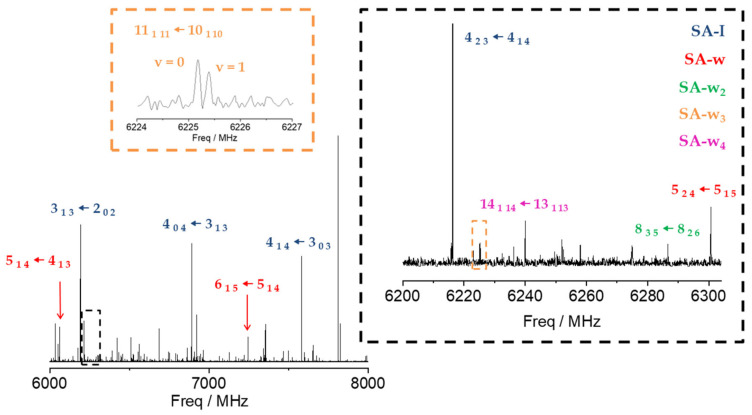
CP-FTMW, 6–8 GHz, rotational spectrum of SA and its water complexes. A selection of the most intense transitions has been pointed out for the SA-I (blue), SA-w (red), SA-w_2_ (green), SA-w_3_ (orange), and SA-w_4_ (purple) species to compare their relative intensities. The excerpt in black facilitates comparison between transitions of all the species observed (SA, SA-w, SA-w_2_, SA-w_3_, SA-w_4_). The excerpt in orange shows the doublets observed for the SA-w_3_ species.

**Figure 3 ijms-25-04074-f003:**
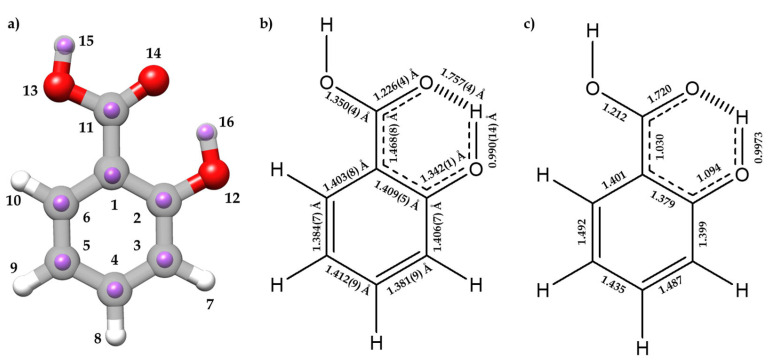
(**a**) The experimental *r*_s_ positions of the isotopically substituted atoms (purple) for I conformer of SA with the *r*_e_ predicted structure at B3LYP-D3/6-311++G(d,p) level of theory. (**b**) and (**c**) are schematic representations showing the resonant form stabilized by the AGIBA effect with (**b**) *r*_m_ bond lengths experimentally determined and (**c**) predicted bond orders from the NBO calculations done at B3LYP-D3/6-311++G(d,p) level of theory.

**Figure 4 ijms-25-04074-f004:**
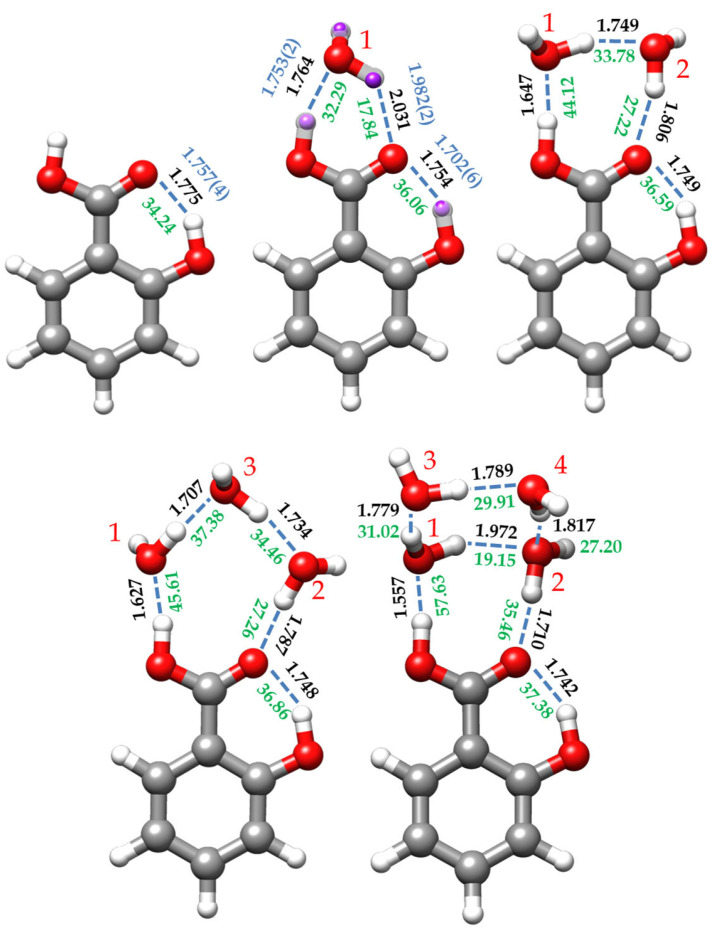
Predicted B3LYP-D3/6-311++G(d,p) *r*_e_ structures, estimated interaction energies (green, in kJ/mol), and hydrogen bond distances (black (theoretical) and blue (experimental), in Å) for the salicylic acid monomer (SA) and its complexes with one (I-w-a), two (I-w_2_-a), three (I-w_3_-a), and four (I-w_4_-a) water molecules. For the SA monomer, the experimental distances correspond to the mass-dependent *r*_m_ (blue) structure. For the monohydrated conformer, SA-w, the *r*_e_ structure is compared with the experimentally determined substitution *r*_s_ (purple) and effective *r*_0_ (blue) structures. The red numbers are the labelling of the water molecules. The interaction energies were estimated from the empirical equation taken from the QTAIM calculations and reference [63] (see text).

**Table 1 ijms-25-04074-t001:** Experimental rotational parameters determined for SA, compared with those predicted for conformer SA-I at B3LYP-D3/6-311++G(d,p) level of theory.

Fitted Param. ^1^	(MW)	(MW + MMW) ^2^	(MMW, ref. [11]) ^3^	SA-I—Theor.
*A*/MHz	2340.25075(42) ^4^	2340.25105(17)	2340.248(2)	2336.30
*B*/MHz	1223.76931(31)	1223.76941(13)	1223.79(2)	1221.78
*C*/MHz	803.89784(31)	803.89788(18)	803.90(1)	802.24
*P*_cc_/uÅ^2^	0.12962(19)	0.12960(10)	0.127	0.00
Δ_J_/kHz	0.1091(90)	0.11722(67)	0.049(6)	0.1897
Δ_JK_/kHz	−0.230(28)	−0.2536(26)	0.130(9)	−0.3939
Δ_K_/kHz	0.134(20)	0.1505(44)	-	0.2128
*δ*_J_/kHz	−0.0303(44)	−0.03392(37)	-	0.0843
*δ*_K_/kHz	0.069(14)	0.0812(18)	-	−0.1244
N	64	114	50	-
*σ*/kHz	6.8	11.1	23.0	-

^1^ *A*, *B*, and *C* are the rotational constants. *P*_cc_ is the planar moment of inertia, derived from *P*_cc_ = (*I*_a_ + *I*_b_ − *I*_c_)/2. Δ*_J_*, Δ*_JK_,* Δ_K_, *δ*_J_, and *δ*_K_ are the quartic centrifugal distortion constants. N is the number of rotational transitions fitted. *Σ* is the rms deviations of the fit. ^2^ Fit of the 64 measured in this work (MW) and the 50 transitions measured in ref. [11] (MMW) transitions observed for SA-I. ^3^ Experimental parameters reported in reference [11]. ^4^ Standard error is given in parentheses in units of the last digit.

**Table 2 ijms-25-04074-t002:** Experimental rotational parameters obtained for the SA-w monohydrated and SA-w_2_ dihydrated complexes, compared with the predicted parameters of I-w-a, II-w-a, I-w_2_-a, and II-w_2_-a conformers of SA–water_n_ (n = 1, 2) complexes calculated at B3LYP-D3/6-311++G(d,p) level of theory.

Fitted Param. ^1^	SA-w	I-w-a	II-w-a	SA-w_2_	I-w_2_-a	II-w_2_-a
*A*/MHz	2315.34456(45) ^2^	2313.15	2311.93	1769.91699(48)	1780.40	1759.99
*B*/MHz	645.08107(14)	648.76	647.51	451.04489(18)	453.67	450.32
*C*/MHz	505.03872(14)	507.09	506.29	360.15478(16)	362.04	359.17
*P*_cc_/uÅ ^2^	0.51743(24)	0.42	0.44	1.38688(57)	0.96	1.17
Δ_J_/kHz	0.0845(37)	0.017	0.016	0.0763(27)	0.008	0.007
Δ_JK_/kHz	−0.0757(32)	0.106	0.104	−0.0699(26)	0.064	0.092
Δ_K_/kHz	-	0.066	0.056	-	0.072	0.072
*δ*_J_/kHz	−0.0274(15)	0.004	0.004	−0.0333(12)	0.001	0.001
*δ*_K_/kHz	-	0.068	0.066	-	0.049	0.049
*µ*_a_/*µ*_b_/*µ*_c_/D	+/+/-	0.6/1.0/1.2	0.7/0.6/1.2	+/+/-	0.5/0.9/0.0	0.8/0.8/0.0
Δ*E*_DFT_/cm^−1^	-	0.0 ^3^	890.6	-	0.0 ^4^	639.9
N	84	-	-	126	-	-
*σ*/kHz	5.4	-	-	7.7	-	-

^1^ *A*, *B*, and *C* are the rotational constants. *P*_cc_ is the planar moment of inertia, derived from *P*_cc_ = (*I*_a_ + *I*_b_ − *I*_c_)/2. Δ_J_, Δ_JK_, Δ_K_, *δ*_J_ and *δ*_K_ are the quartic centrifugal distortion constants. *µ*_a_, *µ*_b_, and *µ*_c_ are the components of the electric dipole moment, + or – mean the observation or not of the corresponding selection rules. Δ*E*_DFT_ is the energy relative to the most stable conformer (I-w-a and I-w_2_-a, respectively). N is the number of rotational transitions fitted. *σ* is the rms deviations of the fit. ^2^ Standard error is given in parentheses in units of the last digit. ^3^ Absolute energy is −572.6949094 Eh. ^4^ Absolute energy is −649.1747737 Eh.

**Table 3 ijms-25-04074-t003:** Experimental rotational parameters obtained for the two vibrational states (ν = 0 and 1) observed for the SA-w_3_ species of the trihydrated complex compared with the predicted parameters of SA-I-w_3_-a, SA-I-w_3_-b, and SA-I-w_3_-c conformers of the SA trihydrated complex calculated at B3LYP-D3/6-311++G(d,p) level of theory.

Fitted Param. ^1^	SA-w_3_ (ν = 0)	SA-w_3_ (ν = 1)	I-w_3_-a	I-w_3_-b	I-w_3_-c
*A*/MHz	1412.46036(98) ^2^	1412.42970(98)	1412.10	1436.05	1421.44
*B*/MHz	315.84360(19)	315.84610(22)	319.03	317.16	327.39
*C*/MHz	260.77801(14)	260.78823(15)	263.04	262.98	284.18
*P*_cc_/uÅ^2^	9.9634(11)	9.9989(12)	10.33	11.82	60.41
Δ_J_/kHz	0.01785(63)		0.018	0.016	0.029
Δ_JK_/kHz	−0.1706(86)		−0.091	−0.077	−0.163
Δ_K_/kHz	1.431(79)		0.846	0.812	1.225
*d*_1_/kHz	−0.00297(44)		−0.001	−0.002	−0.001
*d*_2_/kHz	0.00197(39)		−0.001	−0.001	0.001
*F*_bc_/MHz	0.1304(52)		-	-	-
Δ*E*_01_/MHz	717.0(5.2)		-	-	-
*µ*_a_/*µ*_b_/*µ*_c_/D	+/+/−		0.6/0.6/0.6	0.1/0.0/0.3	1.5/1.3/0.1
Δ*E*_DFT_/cm^−1^	-		0.0 ^3^	130.4	130.4
N	160		-	-	-
*σ*/kHz	7.0		-	-	-

^1^ *A*, *B,* and *C* are the rotational constants. *P*_cc_ is the planar moment of inertia, derived from *P*_cc_ = (*I*_a_ + *I*_b_ – *I*_c_)/2. Δ_J_, Δ_JK_, Δ_K_, *d*_1_, and *d*_2_ are the quartic centrifugal distortion constants. *F*_bc_ is the Coriolis coupling constant. Δ*E*_01_ is the energy difference between the two vibrational states. *µ*_a_, *µ*_b_, and *µ*_c_ are the components of the electric dipole moment, + or – mean the observation or not of the corresponding selection rules. Δ*E*_DFT_ is the energy relative to the most stable I-w_3_-a conformer. N is the number of rotational transitions fitted. *σ* is the rms deviations of the fit. ^2^ Standard error is given in parentheses in units of the last digits. ^3^ Absolute energy is −725.6502119 Eh.

**Table 4 ijms-25-04074-t004:** Experimental rotational parameters obtained for the SA-w_4_ species of the tetrahydrated complex of SA compared with the predicted parameters of I-w_4_-a, I-w_4_-b, I-w_4_-c, and I-w_4_-d conformers of SA tetrahydrated complex calculated at B3LYP-D3/6-311++G(d,p) level of theory.

Fitted Param. ^1^	SA-w_4_	I-w_4_-a	I-w_4_-b	I-w_4_-c	I-w_4_-d
*A*/MHz	1055.8085(24) ^2^	1073.10	1066.29	1059.87	1093.11
*B*/MHz	248.19262(15)	250.60	251.56	251.90	343.76
*C*/MHz	220.88132(19)	222.99	224..02	224.40	217.21
*P*_cc_/uÅ ^2^	113.4455(21)	110.68	113.23	115.48	104.46
Δ_J_/kHz	0.191(46)	0.035	0.035	0.040	0.036
Δ_JK_/kHz	0.094(10)	−0.212	−0.150	−0.175	−0.186
Δ_K_/kHz	−0.239(36)	1.297	1.508	1.332	1.985
*d*_1_/kHz	0.208(44)	0.001	0.001	0.001	0.001
*d*_2_/kHz	−0.135(21)	0.001	0.001	0.001	0.001
*H*_J_/Hz	0.0106(20)	-	-	-	-
*µ*_a_/*µ*_b_/*µ*_c_/D	+/+/-	0.8/1.4/0.2	0.7/1.6/0.4	0.8/1.3/1.6	0.6/3.1/0.5
Δ*E*_DFT_/cm^−1^	-	0.0 ^3^	165.7	241.2	267.8
N	169	-	-	-	-
*σ*/kHz	6.4	-	-	-	-

^1^ *A*, *B,* and *C* are the rotational constants. *P*_cc_ is the planar moment of inertia, derived from *P*_cc_ = (*I*_a_ + *I*_b_ – *I*_c_)/2. Δ_J_, Δ_JK_, Δ_K_, *d*_1_, and *d*_2_ are the quartic centrifugal distortion constants and *H*_J_ is the sextic centrifugal distortion constant. *µ*_a_, *µ*_b_, and *µ*_c_ are the components of the electric dipole moment, + or – mean the observation or not of the corresponding selection rules. Δ*E*_DFT_ is the energy relative to the most stable I-w_4_-a conformer. N is the number of rotational transitions fitted. *σ* is the rms deviations of the fit. ^2^ Standard error is given in parentheses in units of the last digit. ^3^ Absolute energy is −802.1296231 Eh.

**Table 5 ijms-25-04074-t005:** Predicted O⋯O distances between the oxygen atoms forming hydrogen bonds for the identified conformers of SA and its complexes SA-w (I-w-a), SA-w_2_ (I-w_2_-a), SA-w_3_ (I-w_3_-a), and SA-w_4_ (I-w_4_-a), calculated at B3LYP-D3/6-311++G(d,p) level of theory. See Figure 3 for the atom labelling and Figure 4 for the labelling of the water molecules.

O⋯O/Å	SA	I-w-a	I-w_2_-a	I-w_3_-a	I-w_4_-a
O_12_⋯O_14_	2.635	2.620	2.617	2.617	2.612
O_13_⋯O_w1_	-	2.710	2.648	2.616	2.576
O_14_⋯O_w2_	-	2.779	2.766	2.764	2.694
O_w1_⋯O_w2_	-	-	2.682	-	2.835
O_w1_⋯O_w3_	-	-	-	2.691	2.740
O_w2_⋯O_w3_	-	-	-	2.698	-
O_w3_⋯O_w4_	-	-	-	-	2.745
O_w2_⋯O_w4_	-	-	-	-	2.781
O⋯O w-dimer ^1^	2.976				
O⋯O w-trimer ^2^	2.85				
O⋯O w-tetramer ^3^	2.78				

^1^ Taken from reference [64]. ^2^ Taken from reference [66]. ^3^ Taken from reference [65].

**Table 6 ijms-25-04074-t006:** Predicted equilibrium dissociation energies dissociation energies *D*_e_ and average dissociation energies per intermolecular bond (*D*_e/HB_) for the identified conformers of salicylic acid—water complexes, SA-w (I-w-a), SA-w_2_ (I-w_2_-a), SA-w_3_ (I-w_3_-a), and SA-w_4_ (I-w_4_-a), calculated at MP2/6-311++G(2df,2dp) level of theory and employing the counterpoise method to correct the basis set superposition error (BSSE).

	I-w-a	I-w_2_-a	I-w_3_-a	I-w_4_-a
*D*_e_/kJ mol^−1^	41.34	88.70	123.34	168.87
*D*_e/HB_/kJ mol^−1^	20.67	29.57	30.84	28.14

## Data Availability

The data that support the findings of this study are available in the Appendix A of this article.

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
