# Peer review of "Molecular Structure of Salicylic Acid and Its Hydrates: A Rotational Spectroscopy Study"

_ijms, 2024, doi:10.3390/ijms25074074_

Round 1

Reviewer 1 Report

Comments and Suggestions for Authors

The manuscript entitled “Molecular Structure of Salicylic Acid and its Hydrates: A Rotational Spectroscopy Study” submitted by S Blanco and coworker presents a combined experimental and theoretical investigation about the salicylic acid – water cluster structure (up to four water molecule) by using chirped pulse Fourier transform microwave spectroscopy and ab initio molecular modeling techniques. The research topic explored in the manuscript looks of having a significant importance since it is a systematic and exhaustive investigation about the nature of the bonding between different monomeric forms inside the cluster structure. The manuscript is well written and contains an interesting methodological discussion and analysis. Both experimental and molecular modelling techniques have been applied at the highest scientific level, and the results obtained with these methods are of great interest

General observation:

-          Is it possible to compare the pure water structures with those found for salicylic acid – water cluster. For ex. SA-(H2O)4 with (H2O)6 or SA-(H2O)3 with (H2O)5 or SA-(H2O)2 with (H2O)4? See Phys. Chem. Chem. Phys., 2021, 23, 18734-18743, DOI: 10.1039/D1CP02301B

The manuscript is suitable for publication in the IJMS journal, even at the present form, but if the Authors are willing to add a brief comment on my observation, I would appreciate it.

Author Response

We would like to thank the reviewer for their comments, which have improved the content of the article.

We have made the changes in the manuscript, highlighting them in yellow.

Below, we address point by point the corresponding comments:

Referee comment:

          Is it possible to compare the pure water structures with those found for salicylic acid – water cluster. For ex. SA-(H2O)4 with (H2O)6 or SA-(H2O)3 with (H2O)5 or SA-(H2O)2 with (H2O)4? See Phys. Chem. Chem. Phys., 2021, 23, 18734-18743, DOI: 10.1039/D1CP02301B

Author reply:

We already have done a brief comparison between the water clusters and the SA water complexes including some data in Table 5. I addition, in page 16 we have incorporated the following sentence:

“In the work on formamide-(H2-O)3 complex [28] the structural relationship between the formamide-(H2O)n clusters and the pure water clusters (H2O)n+2 was pointed out. The structures determined in this work for the SA-wn complexes could be related in the same way to the structures determined or predicted for the (H2O)n+2 clusters [20,64,65,66,84].”

incorporating the reference suggested (ref. 84).

Reviewer 2 Report

Comments and Suggestions for Authors

This study used microwave rotational spectroscopy to analyze the molecular structure and hydrogen bonding of salicylic acid and its hydrates with up to four water molecules. The authors used theoretical simulation (DFT) to uncover how water molecules cluster around the acid group and the cooperative effects enhancing hydrogen bonding. This is a detailed work with enough supporting information. This draft can be accepted after a necessary discussion to determine whether the authors also calculated the structure with water molecules combining with the phenol group.

Author Response

We would like to thank the reviewer for their comments, which have improved the content of the article.

We have made the changes in the manuscript, highlighting them in yellow.

Below, we address point by point the corresponding comments:

Referee comment:

This draft can be accepted after a necessary discussion to determine whether the authors also calculated the structure with water molecules combining with the phenol group.

Author reply:

We have incorporated the following sentences

in page 5:

Some of the resulting structures are given in Figures S4-S6 and the spectroscopic parameters in Tables S3-S5. From the predicted energies it can be deduced that the predicted preferred interaction sites are related to the COOH group. As examples we have included some of the possible complexes with water molecules closing rings with the phenol group that have in all cases rather high energies relative to the global minima”

in page 15:

As we have already mentioned the possible aggregates around the phenolic OH group have all high energies (see Figures S2-S5). In the complex phenol-w the O-H group acts preferably as proton donor, form predicted to be much more stable than the form in which water is the proton donor [83]. However, in the most stable forms of SA the proton donor capacity of the phenolic OH group is employed in the intramolecular interaction so the possible SA- water complexes with water interacting with the phenol O-H group are expected to have higher energies as confirmed by the calculations.

Reviewer 3 Report

Comments and Suggestions for Authors

This study deals with salicylic acid and its hydrates with up to four water molecules by means of chirped pulse Fourier transform microwave spectroscopy. Moreover, a computational technique was presented and model predictions were validated by comparison with experimental data.

The whole work is interesting.

Points for improvement:

1. A literature review by using both GoogleScholar and authors' keywords revealed about 17,000 works Please, kindly complete literature and/or keywords.

2. Please, describe in the introduction the general status in the area, specify the task of this work and highlight your contributions.

3. Please, provide as suplementary material detailed data for the parameters used in simulations

4. Please, compare the experimental and theoretical estimated data with other colleagues data for similar cases.

5. Please, describe shortly the principles of the used software,

Author Response

We would like to thank the reviewer for their comments, which have improved the content of the article.

We have made the changes in the manuscript, highlighting them in yellow.

Below, we address point by point the corresponding comments:

Referee comments:

  1. A literature review by using both GoogleScholar and authors' keywords revealed about 17,000 works Please, kindly complete literature and/or keywords.

Author reply:

It is true that there are numerous references on complexes with water. We did not intend to conduct a review on water complexes or exceed the number of references in the paper. However, the reviewer's comment is right, and we take this opportunity to include new references on more recent water complexes that present significant results. We have included references 17, 20, 22, 23, 24, 25, 27, and 33 in the introduction, to which we have also referred in some cases throughout the manuscript.

  1. Please, describe in the introduction the general status in the area, specify the task of this work and highlight your contributions.

Author reply:

Taking advantage of the inclusion of the references, we have added the following paragraph to pages 2 and 3 in the introduction:

The investigation of microsolvated molecular systems is a relevant subject for Chemistry and biochemistry [17,18,19]. Microwave spectroscopy techniques are exceptional tools to determine the structure, which are adequate for this purpose due to their inherent high resolution [20]. There is a high number of microwave studies of microsolvated organic molecules [19,21,22,23,24,25] providing models for a better understanding of the water interactions with biomolecules [26], of hydrogen bond (HB) cooperativity [27] and of the way in which solvation induces structural changes in the solute molecule [22,23,27,28]. These studies have also led to understand the role of association processes and the interplay between self-association of water and solvation. With few exceptions [22], in complexes with several H2O molecules water prefers to link other water molecules forming chains or cycles. When the solute has only one HB acceptor site the structures reflect a balance between maximizing the number of water-solute interactions, and the minimum-energy structures of the pure water clusters [21]. In solutes with double HB donor/acceptor character, water molecules close sequential cycles as observed in amides [28], acids [29,30,31,32] or esters [33]. In the last years, studies of molecules forming clusters with a high number of water molecules has increased.

With this paragraph and next paragraphs in the introduction, we think that the task of this work and our contributions are better explained.

  1. Please, provide as suplementary material detailed data for the parameters used in simulations

Author reply:

We do not understand to which simulations do refer.

The method and basis set used to optimize the structures and calculate the rotational parameters are described in the methods. The rotational constants, dipole moments and values of the relative energies that we have used to predict the spectra are in the corresponding tables.

To our knowledge, we have not reported any simulation.

  1. Please, compare the experimental and theoretical estimated data with other colleagues data for similar cases.

Author reply:

We have incorporated the following sentences in page 16:

The cyclic structures of the hydrates of SA are comparable with those of reported by Howard and coworkers on organic acid mono, di and trihydrates [29,30,31,32]. However, as a difference with the non-planar heavy atom skeleton observed here for the trihydrate of SA, the trihydrates of di and trifluoroacetic acids seems to have nearly planar heavy atom skeletons.

In the work on formamide-(H2-O)3 complex [28] the structural relationship between the formamide-(H2O)n clusters and the pure water clusters (H2O)n+2 was pointed out. The structures determined in this work for the SA-wn complexes could be related in the same way to the structures determined or predicted for the (H2O)n+2 clusters [20,64,65,66,84].

  1. Please, describe shortly the principles of the used software,

Author reply:

We do not understand the question, since all the packages used are described in the material and methods section with the corresponding references.

Round 2

Reviewer 3 Report

Comments and Suggestions for Authors

The authors promptly responded to my review report. The manuscript is accepted